# DELVE INTO IMAGE STYLE DIFFUSION TOWARDS SCHRÖDINGER BRIDGE PROBLEM

## ABSTRACT

Taking inspiration from the exceptional performances of Score-Based Generative Modeling (SGM) in image generation tasks, we introduce a novel Style-Diffusion method in this work. For the first time, we achieve flexible and efficient stylization transfer using SGM while preserving the semantic structures. With the prior distributions $p_\theta(v)$ obtained from encoding the source domain data samples, we employ approximate score-matching to estimate the drift of the reverse-time Stochastic Differential Equation (SDE) at arbitrary time step. By introducing Control Factor $\phi$, we have achieved controllable stylization in the output images. To improve computation speed, we re-formulate the original multi-end diffusion problem as a composite Schrödinger half bridge Problem, providing a new method for the diffusion evolution between more complex multiple distributions. Numerous empirical results and comparison with state-of-the-art methods demonstrate the superior performance of our approach in terms of stylization and extraordinary preservation of semantic structure.

## 1 INTRODUCTION

Score-based Generative Modeling (SGM) is a novel approach for probabilistic generative modeling that has demonstrated state-of-the-art performance on various audio and image synthesis tasks (Cai et al., 2020; Kong et al., 2020; Jolicoeur-Martineau et al., 2020; Song et al., 2020a; Saharia et al., 2022b; Popov et al., 2021). GLIDE (Nichol et al., 2021), DALL-E 2 by OpenAI (Ramesh et al., 2022), and Imagen by Google (Saharia et al., 2022a) are among the successful examples of SGMs. In the field of image generation, many recent outstanding works are also based on diffusion model. Poole *et.al* (Poole et al., 2022) proposed DreamFusion, which utilizes a 2D diffusion model as a prior for optimizing the parameterized image generator, to generate 3D models from given text that can be viewed from any angle. Rombach *et.al* (Rombach et al., 2022) proposed Stable Diffusion (Latent Diffusion Model), which introduced cross-attention layers to transform the diffusion model into a powerful and flexible generator for high-resolution synthesis. Building on the former, Takagi *et.al* (Takagi & Nishimoto, 2022) achieved the reconstruction of high-resolution images from high-fidelity brain activity from functional Magnetic Resonance Imaging (fMRI) signals. Recent works demonstrate that diffusion models have advanced to provide state-of-the-art performance (Dhariwal & Nichol, 2021).

SGM, inspired by non-equilibrium thermodynamics, typically consists of two parts. In the forward process, data is perturbed by gradually adding noise, leading to a prior distribution that is easy to sample from. In the reverse process, a generative model is constructed by using a trainable neural network to predict noise that depends on the time index $t$ (Ho et al., 2020; Song & Ermon, 2019b;a). Song *et.al* (Song et al., 2020b) presented a comprehensive framework and provided a SDE representation of the noise process. The challenging issue faced by SGM is the high computational cost, and training for a sufficiently long time is necessary to approximate the prior distribution (De Bortoli et al., 2021). Furthermore, in order to ensure that the noise prediction in the reverse process also conforms to a simple distribution *e.g.* Gaussian distribution, the variance *variance schedule* $\beta_t$ should be set small enough. To alleviate this issue, Bortoli *et.al* (De Bortoli et al., 2021) proposed Diffusion Schrödinger Bridge (DSB), which formulates the generative modeling process of SGM as an SBP problem, *i.e.* an entropy-regularized optimal transport (OT) problem (Schrödinger, 1932) on path spaces. Experimental results show that with this approach can effectively reduce diffusion steps while ensuring the quality of generated samples.

Convolutional neural networks (CNNs) have always been a popular solution for generative models (Gatys et al., 2016; 2017), as they are able to extract both style and content from pre-trained networks in order to optimize the joint content and style loss of pending images. Additionally, various CNN-based style transfer methods have been proposed (Kolkin et al., 2019; Wang et al., 2020; Kalischek et al., 2021; Chen et al., 2021; Hong et al., 2021; Zhu et al., 2017), which are highly efficient for feature extraction in source images. However, To the best of the author's knowledge, there is currently no work that utilizes a diffusion-based model for style transfer, which allows for flexible and efficient stylization transfer while preserving the semantic structure. Inspired by the superior generative performance of SGM, we have, in this work, for the first time, incorporated diffusion model into the domain of I2I style transfer.

In this work, we propose a new Style-Diffusion method for the gradual evolution of content and style image distributions. In contrast to the traditional SGM method, our model obtains the desired output at the midst layer. Instead of using the U-net architecture (Ronneberger et al., 2015) to obtain a noise predictor, $\varepsilon_t^\theta$, we use a CNN with parameter $\theta$ as an encoder to extract style and content feature targets as prior distribution from both domains, denoted as $p_\theta(v)$. We use $p_\theta(v)$ as reference to obtain approximate score-matching for arbitrary sampled data, and use it to estimate the time inhomogeneous drift of the corresponding reverse process SDE. Furthermore, by introducing the control factor $\phi$, we enable control over the degree of stylization in the output images, allowing for different degrees of stylization to be output according to the specific needs and requirements of different application scenarios and tasks. For instance, when emphasizing the semantic content structure, we can increase $\phi$, whereas when emphasizing the original artistic style of the style image (*e.g.* by choosing an abstract style image, people may prefer stronger symbolic features in the output image), we can decrease the value to give the image stronger stylistic characteristics. Notably, Style-Diffusion exhibits a remarkable capability in retaining semantic structures when compared to other state-of-the-art methods, see Appendix. To reduce computational costs and minimize the number of iterations, we reformulate the diffusion process as a Schrödinger half bridge composite problem and provide optimization objectives.

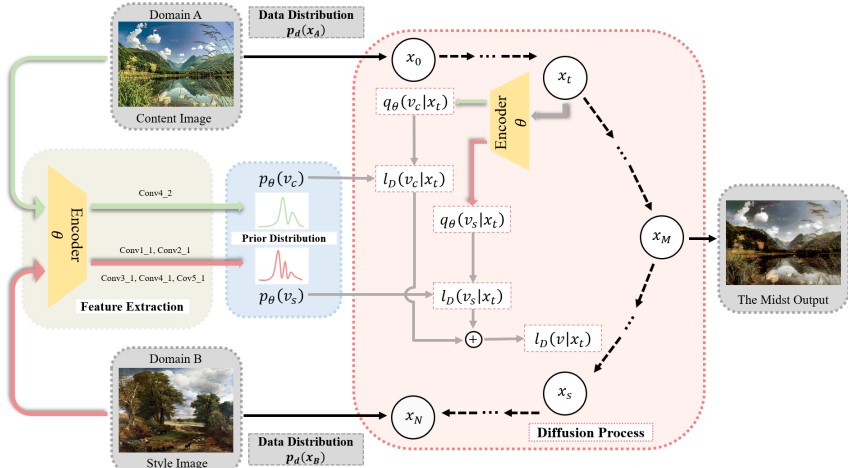

Figure 1: The overall framework of our proposed Style-Diffusion method. To implement feature extraction, we utilize the VGG encoder with parameter $\theta$. The priors $p_\theta(v_c)$ and $p_\theta(v_s)$ serve as references to guide data construction. For an arbitrary $x(t)$ that depends on time $t$, the encoder can generate corresponding sample feature distributions $q_\theta(v_c|x_t)$ and $q_\theta(v_s|x_t)$. The distance between the distribution at time $t$ and the prior reference is denoted as $l_D$. In sections 3.3 and Appendix E, we provide a detailed illustration and experimental analysis of the distance measurement method.

In summary, our substantial contributions includes:

- We propose a novel controllable Style-Diffusion method, which to the best of our knowledge, is the first diffusion-based method for targeted I2I style transfer.

- We are the first to give a mathematical formulation for the multi-end diffusion task under the perspective of the Schrödinger bridge problem (SBP).

- We present comprehensive experimental results and detailed comparisons with state-of-the-art methods (Deng et al., 2022; Wu et al., 2021; Liu et al., 2021; Park & Lee, 2019; Huang & Belongie, 2017), showing that our method outperforms baseline methods and demonstrates exceptional ability in preserving semantic structures (see Appendix D).

## 2 RELATED WORKS

### 2.1 SCORE-BASED GENERATIVE MODELS (SGMS)

Score Matching with Langevin dynamics (SMLD) (Song & Ermon, 2019a) and Denoising Diffusion Probabilistic Models (DDPM) (Ho et al., 2020) are two classes well-known probabilistic generative models, Song *et.al* (Song et al., 2020b) refers to these two models as score-based generative models and proposes a comprehensive framework that extends prior methods by examining them through the perspective of stochastic differential equations (SDEs). According to their work, the diffusion process can be expressed as the solution to an Itô SDE:

$$dx = f(x,t)dt + g(t)dw \tag{1}$$

where $w$ is the standard Wiener process (a.k.a., Brownian motion), $f(\cdot, t) : \mathbb{R}^d \to \mathbb{R}^d$ is the *drift* coefficient of $x(t)$, and $g(\cdot) : \mathbb{R} \to \mathbb{R}$ is the *diffusion* coefficient of $x(t)$. The reverse process is also a diffusion process (Anderson, 1982), and can be written by the reverse-time SDE:

$$dx = [f(x,t)dt - g(t)^2 \nabla_x \log p_t(x)]dt + g(t)d\bar{w} \tag{2}$$

where $\bar{w}$ is a standard Wiener process in backwards flow, $dt$ is an infinitesimal negative timestep, and $\nabla_x \log p_t(x)$ is the score function of each marginal distribution.

To estimate $\nabla_x \log p_t(x)$, Song *et.al* (Song et al., 2020b) train a score-based model $s_\theta(x,t)$ with parameter $\theta$:

$$\theta^* = \arg\min_\theta \mathbb{E}_t \{\lambda(t)\mathbb{E}_{x(0)}\mathbb{E}_{x(t)|x(0)}[\| s_\theta(x(t),t) \\ - \nabla_{x(t)} \log p_{0t}(x(t)|x(0))\|_2^2]\} \tag{3}$$

where $\lambda : [0,T] \to \mathbb{R}_{>0}$ is a positive weighting function, $t$ is uniformly sampled over $[0,T]$. Though training the time-dependent score-based model $s_\theta(x,t)$, the reverse-time SDE can be constructed to generate samples from $p_0$.

**Probability Flow ODE.** An alternative approach to solving the reverse-time stochastic differential equation (SDE) exists. Song *et.al* (Song et al., 2020b) propose that a deterministic ordinary differential equation (ODE) can be used to represent all diffusion processes, which shares the same marginal densities as the corresponding SDE. The ODE is expressed as follows:

$$dx = [f(x,t) - \frac{1}{2}g(t)^2 \nabla_x \log p_t(x)]dt \tag{4}$$

which is named as *probability flow* ODE.

The traditional diffusion model (as shown in Figure 2) is inspired by the non-equilibrium thermodynamic entropy increase, where noise is continuously added to a given input data in the forward process, ultimately resulting in isotropic Gaussian noise. Through the following discussion of SBP, we will find that SGMs are a special case of the SBP, where the data distributions at both ends of SBP can be more complex, rather than easily samplable distributions such as *Gaussian*. Furthermore, this end-to-end diffusion model cannot effectively handle the diffusion process between multiple distributions.

### 2.2 SCHRÖDINGER BRIDGE PROBLEMS (SBPS)

The Schrödinger bridge problem (SBP) seeks to find optimal stochastic evolution between two probability distributions, given a prior or reference (Vargas, 2021).

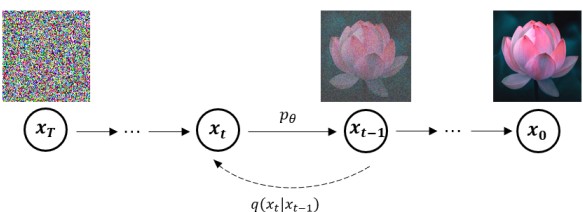

Figure 2: The end-to-end structure of classic diffusion model

**Definition 1** (Dynamic Schrödinger Problem). The dynamic Schrödinger problem is given by

$$\inf_{\mathbb{Q}\in\mathcal{D}(\alpha_0,\alpha_1)} D_{KL}(\mathbb{Q}\|\mathbb{W}^\delta) \tag{5}$$

where $\mathcal{D}(\alpha_0,\alpha_1)$ represents the set of path measures with marginals $\alpha_0$ and $\alpha_1$ at times 0, 1, and $\mathbb{W}^\delta$ is the Wiener measure with volatility $\delta$.

**Static Schrödinger Bridge.** Föllmer (Föllmer, 1988) gives the optimal density solution $q^*(x,y)$ of the static Schrödinger Bridge under the marginal constraints:

$$q^*(x,y) = \arg\inf_{q(x,y)} D_{KL}(q(x,y)\|p^{\mathbb{W}^\delta}(x,y))$$

$$s.t. \ \alpha_0(x) = \int q(x,y)dy, \alpha_1(y) = \int q(x,y)dx$$

It is evident that the extremization of the exponential function in Equation (5) can present a challenging numerical problem due to the simultaneous satisfaction of two boundary value constraints.

**Half Bridge.** To simplify the original full bridge problem, Pavon *et.al* (Pavon et al., 2021) proposed the half-bridge problem, which removes one of the constraints and integrate it as an initial value problem.

**Definition 2.** The forward half bridge is given by

$$\mathbb{Q}^* = \inf_{\mathbb{Q}\in\mathcal{D}(\alpha_0,\cdot)} D_{KL}(\mathbb{Q}\|\mathbb{W}^\delta) \tag{6}$$

**Definition 3.** The backward half bridge is given by

$$\mathbb{P}^* = \inf_{\mathbb{P}\in\mathcal{D}(\cdot,\alpha_1)} D_{KL}(\mathbb{P}\|\mathbb{W}^\delta) \tag{7}$$

The resolution of the half bridge problem is a significant contribution towards constructing an iterative approach for solving the full bridge problem. In the Section 3, we formulate the task of SGMs style transfer as a composite SBP. In this way, we can improve computational speed and provide a feasible method for multi-end

## 3 METHODS

### 3.1 STYLE AND CONTENT FEATURE VECTORS

Existing diffusion-based methods leverage the U-Net architecture (Ronneberger et al., 2015) to obtain a noise predictor, $\varepsilon_t^\theta$. Different from the noise adding way of traditional diffusion model, which utilizes the Gaussian noise during the forward process. We impose the target vectors to guide the noise in each unique time step. Inspired by the work (Gatys et al., 2016), which introduces A Neural Algorithm of Artistic Style that can separate and recombine the image content and style of natural images by utilizing the convolutional neural network (CNN) as an encoder, we use a pre-trained CNN to extract 5 style feature target vectors $(s_1, s_2, s_3, s_4, s_5)$ and 1 content feature target vector $(c_1)$ from two domains, and then impose these feature targets to guide the noise, $\varepsilon_t^\theta(x_t|\sum_{i=1}^5 s_i, c_1)$.

## 3.2 ALGORITHM OF OUR METHOD

Given $x_A$ and $x_B$ are samples drawn from the true data distribution, $p_d(X_A)$ and $p_d(X_B)$, respectively, belonging to domain A ($\mathcal{X}_A$) and domain B ($\mathcal{X}_B$). We continuously add noise over $0 \sim T$ time steps. $x_1, ..., x_t$ denote the samples in each step. Our objective is to find a distribution evolution path from $p_d(X_A)$ to $p_d(X_B)$, and the midst distribution $q(x_M|x_A)$ combines the feature constraints from both domains.

Let $v_s, v_c$ be the convolutional layer style and content feature vectors extracted by utilizing a CNN as an encoder and $p_\theta(v)$ be the prior distribution we want to impose on the codes with CNN parameters, $\theta$. The conditional style and content feature distribution can be obtained as $p_\theta(v_c|x_A)$ and $p_\theta(v_s|x_B)$. The aggregated distribution of $p_\theta(v_c)$ and $p_\theta(v_s)$ can be obtained as:

$$p_\theta(v_c) = \int p_\theta(v_c|x_A)p_\theta(x_A)dx_A \tag{8}$$

$$p_\theta(v_s) = \int p_\theta(v_s|x_B)p_\theta(x_B)dx_B \tag{9}$$

which represent the content and style targets codes distribution.

Given $x_t$ sampled from the intermediate steps $x_t \sim q(x_t|x_A)$ in the diffusion path, and $p_\theta(v_c|x_t)$, $p_\theta(v_s|x_t)$ be the encoding distributions, the aggregated distribution of content and style codes of the sampled data can be written as:

$$q_\theta(v_c) = \int\int p_\theta(v_c|x_t)q(x_t|x_A)p_\theta(x_A)dx_tdx_A \tag{10}$$

$$q_\theta(v_s) = \int\int p_\theta(v_s|x_t)q(x_t|x_A)p_\theta(x_A)dx_tdx_A \tag{11}$$

With the diffusion sampling $q(x_t|x_A)$, we sample data $x_M$ from the midst layer of our model. The middle data distribution can then be obtained from $q(x_M|x_A)$. To adjust the content preservation of the midst output results, we introduce the hyperparameter $\phi$, defined as the Control Factor (CF) of semantic structure from the domain $\mathcal{X}_A$. We can then obtain the optimal function for the midst data distribution loss based on the true data distribution $p_d(x_A)$ from $\mathcal{X}_A$ as follows:

$$\mathcal{L}_A = D_{KL}[q(x_M|x_A)\|p_d(x_A)] \tag{12}$$

which is to optimize the output distribution of the midst layer, so that the distribution can jointly embody the features of $\mathcal{X}_A$, thereby enabling the final output image to preserve the semantic structure from the content image.

## 3.3 METRIC SPACE

To combine the dual-domain features and optimal the model, we need to focus on the distance between $q_\theta(\cdot)$ and the prior $p_\theta(\cdot)$. Considering that the stochastic process $\{x(t) : t \in T\}$ is a discrete probability function sequence indexed by time t, $q_\theta(\cdot)$ and $p_\theta(\cdot)$ w.r.t Lebesgue measure both with support on $\mathbb{R}^n$, we research the following metric spaces for analysis in the experiment.

**Definition 4** (distance of $l^p$ space). *Given two points $x = \{x_k\} \in l^p$ and $y = \{y_k\} \in l^p$, the distance between $x, y$ is given by: $d(x, y) = \left[\sum_{k=1}^{\infty}|x_k - y_k|^p\right]^{\frac{1}{p}}$*

**Definition 5** (distance of $\mathbb{C}$ space). *Consider the set $\mathbb{C}$ consisting of all convergent sequences of real numbers. For any pair of points $x = \{\xi_i\}$ and $y = \{\eta_i\}$ in $\mathbb{C}$, the distance between $x, y$ is given by: $d(x, y) = \sup_i |\xi_i - \eta_i|$*

**Definition 6** (distance of $\mathbb{R}^n$ space). *Given two points $x = \{x_k\}$ and $y = \{y_k\}$ in Euclidean space $\mathbb{R}^n$, their distance is defined as: $d(x, y) = \sum_{k=1}^{\infty}|x_k - y_k|^2$*

Given a distance metric $L_D[\cdot\|\cdot]$, with $p_\theta(v_s)$, $p_\theta(v_c)$, $q_\theta(v_s)$ and $q_\theta(v_c)$ derived from Equation (8) $\sim$ (11), the objective function can be expressed as:

$$\mathcal{L}_{dual} = \sigma \underbrace{L_D[q_\theta(v_c)\|p_\theta(v_c)]}_{content} + \tau \underbrace{L_D[q_\theta(v_s)\|p_\theta(v_s)]}_{style} \tag{13}$$

which joins the content and style targets feature vector. And the first term in the equation represents the texture and style loss in comparison to targets from domain $\mathcal{X}_A$, while the second term represents the content loss in comparison to targets from domain $\mathcal{X}_B$.

Ultimately, with Equation (12) and (13) we can obtain the final loss function for our Style-Diffusion models, as shown below:

$$\mathcal{L}_{total} = \mathcal{L}_{dual} + \phi\mathcal{L}_A \tag{14}$$

where the first term in the equation represents the loss in the feature of the convolutional layers between $[\dot{v}_c \sim p_\theta(v_c|x_t), \dot{v}_s \sim p_\theta(\dot{v}_s|x_t)]$ (with $x_t \sim q(x_t|x_A)$ ) and $[v_c \sim p_\theta(v_c|x_A), v_s \sim p_\theta(v_s|x_B)]$ (with $x_A \sim p_d(X_A)$, $x_B \sim p_d(X_B)$) that are extracted by our model. The second and third terms denote the KLD between the true data sampled from the true distributions representing $\mathcal{X}_A$ and $\mathcal{X}_B$ and the data sampled from the middle step of the diffusion model. In our experiments, we optimize the middle output by adjusting the hyperparameters $\phi$ to be as close as possible to the artistic target ($\mathcal{X}_B$) while preserving the semantic structure ($\mathcal{X}_A$).

### 3.4 DIFFERENCE BETWEEN STYLE-DIFFUSION AND SGM

According to the *probability flow* ODE proposed by Song *et.al* (Song et al., 2020b), any diffusion model process can be expressed as Equation (4), which can be determined from the SDE once scores are known. The original work utilizes the trainable score-based model $s_\theta(x, t)$ with parameter $\theta$ to estimate $\nabla_x \log p_t(x)$. In the Style-Diffusion, We use a CNN with parameters $\theta$ as the encoder to obtain the feature vector $v$ from the constructed data distribution that depends on time index $t$. Given any $x(t)$ sampled from marginal distribution, we can obtain its feature vector distribution $q_\theta(v|x_t)$ with time index t. With prior $p_\theta(v)$ and the distance metric $L_D[\cdot\|\cdot]$ illustrated in previous part, we utilize $L_D[p_\theta(v)\|q_\theta(v|x_t)]$ as score function for estimation, which is denoted as $L_{D_t}^\theta(x)$.

**Definition 7.** The *aggregated drift* evolves from finite timestep: $t_1 \rightarrow t_2$ is given by:

$$\underset{t1:t2}{\Delta}(x, t) := \int_{t_1}^{t_2} [f(x, t) - \frac{1}{2}g(t)^2 L_{D_t}^\theta(x)]dt \tag{15}$$

Then the midst output is: $x_M = x(0) + \Delta_{0:N/2}(x)$. In D3PSR, time index $t = 0$ indicates that the data sampled from the Domain A ($x_A \sim p_d(X_A)$).

### 3.5 STYLE-DIFFUSION UNDER SBP REPRESENTATION

In SBP, $\Omega = C([0, 1]; \mathbb{R}^n)$ means the path space representing a function of the form $x : [0, 1] :\rightarrow \mathbb{R}^n$, with marginals $\alpha_0, \alpha_1$ at time $t = 0, t = 1$, respectively. The goal is to find $\mathbb{Q}^*$ such that:

$$\mathbb{Q}^* = \arg\min\{D_{KL}[\mathbb{Q}\|\mathbb{W}] : \alpha_0 = p_{data}, \alpha_1 = p_{prior}\} \tag{16}$$

In the Style-Diffusion with the background of diffusion model, the path space is in the form $x : [0, N] :\rightarrow \mathbb{R}^n$, with $\alpha_0 = p_d(X_A), \alpha_N = p_d(X_B)$ at time $t = 0, t = N$, respectively. **Notably,** instead of setting both sampled data distribution at time $t = 0, t = N$ as marginal constraints, we impose the guided feature targets distribution $p_\theta(v)$ to be one of the prior marginals. Then the marginal constraints in the $1^{st}$ process are: $\alpha_0 = p_d(X_A)$, $\alpha_{N/2} = p_\theta(v)$; the marginal constraints in the $2^{nd}$ process are: $\alpha_{N/2} = p_\theta(v)$, $\alpha_N = p_d(X_B)$.

**The First Process.** In the $1^{st}$ process of our approach, we can rewrite it though the perspective of SBP:

**Problem 1.** *The $1^{st}$ process is to find a distribution from $\mathcal{D}(\alpha_0, \alpha_{N/2})$ that minimizes the KL-divergence:* $\mathbb{Q}^{1*} := \arg\min\{D_{KL}(\mathbb{Q}^1\|\mathbb{W}^1) \mid \mathbb{Q}^1 \in \mathcal{D}(\alpha_0, \alpha_{N/2})\}$, *where* $\alpha_0 = p_d(X_A)$, $\alpha_{N/2} = p_\theta(v)$, $\mathbb{W}^1$ *is a prior reference measure.*

**The Second Process.** Similar to the $1^{st}$ process, the $2^{nd}$ process can also be represented as a description in terms of SBP.

**Problem 2.** *The $2^{nd}$ process is to find a distribution from $\mathcal{D}(\alpha_{N/2}, \alpha_N)$ that minimizes the KL-divergence:* $\mathbb{Q}^{2*} := \arg\min\{D_{KL}(\mathbb{Q}^2\|\mathbb{W}^2) \mid \mathbb{Q}^2 \in \mathcal{D}(\alpha_{N/2}, \alpha_N)\}$, *where* $\alpha_{N/2} = p_\theta(v)$,

$\alpha_N = p_d(X_B)$, $\mathbb{W}^2$ *is a prior reference measure.* (The whole proof and derivations are detailed in Appendix A.)

Based on the aforementioned proofs and derivations, we can give the following proposition:

**Proposition 1.** *By considering an evolution between two source domains $\mathcal{X}_A$ and $\mathcal{X}_B$, we define $\mathcal{D}(\alpha_0, \alpha_N)$ as the set of full path measures with marginals $\alpha_0$ and $\alpha_N$. To decompose the full process into half-bridge problems, we can express the original evolution path as:*

$$\text{forward: } \mathbb{Q}^{1*} + \mathbb{Q}^{2*} \tag{17}$$

$$\text{backward: } \mathbb{P}^{1*} + \mathbb{P}^{2*} \tag{18}$$

*which is separated at the time index $t = N/2$.*

Thus, we have expressed the original multi-end diffusion-based problem as a segmented SBPs problem and provided the objectives. To numerically solve this problem, we can utilize Fortet's Algorithm and Iterative Proportional Fitting (IPF) procedure (Fortet, 1940; Kullback, 1968; Ruschendorf, 1995; Gramer, 2000). More details can be found in Appendix A

## 4 EXPERIMENTS

### 4.1 IMPLEMENTING DETAILS

We train our model with MS-COCO dataset (Lin et al., 2014) as the content dataset and the WikiArt dataset (Phillips & Mackintosh, 2011) as the style dataset, each containing roughly 80,000 images of real photos and artistic images respectively. For each content/style image, we set 512×512 as the default image resolution. The content feature target vector is extracted from layer 'conv4_2', and the style feature target vectors are extracted from layers 'conv1_1', 'conv2_1', 'conv3_1', 'conv4_1' and 'conv5_1'. We set the control factor $\phi = 128$ and the ratio $\sigma/\tau$ is set to the same as (Gatys et al., 2016). Our code is implemented with PyTorch (Paszke et al., 2017) and our model is trained on 3 NVIDIA Tesla V100 GPUs.

### 4.2 UNIFYING STRUCTURE AND TEXTURE SIMILARITY

Traditional metrics such as Mean Absolute Error (MAE) and Multi-Scale Structural Similarity (MS-SSIM) (Wang et al., 2003) as well as SSIM (Wang et al., 2004) were relatively inadequate for analyzing texture similarity between images, as they rely on simple introjection mapping and tend to produce conservative estimates that are a combination of all possible results. In order to achieve robustness to texture details (which do represent the same type of object although the local details are different, *e.g.* different areas of lawn), Ding *et.al* (Ding et al., 2020) proposed *Unifying Structure and Texture Similarity* for evaluating the structural and textural similarity between images.

### 4.3 COMPARISON AGAINST OTHER STATE-OF-THE-ART METHODS

In this study, we conducted a comparison of our approach with several state-of-the-art methods, including StyTr2 (Deng et al., 2022), StyleFormer (Wu et al., 2021), AdaAttN (Liu et al., 2021), SANet (Park & Lee, 2019) and AdaIN (Huang & Belongie, 2017) as depicted in Figure 3. Our analysis was based on the Unifying Structure and Texture Similarity method proposed by Ding *et.al* (Ding et al., 2020). Extra samples can be found in the Appendix F

**Qualitative Evaluation.** Our model excels at preserving the semantic content of Domain A after stylistic rendering. Specifically, in Sample 3, the content image from Domain A has a cloud outline that appears blurred. Although other methods have produced output images that largely lost or distorted the cloud information, our model is still capable of rendering the clouds relatively well. This serves as a strong indication of the accuracy of our model in capturing and preserving the semantic content contours. Additionally, in Sample 1, the stylized images from Domain A depict a dreamy night scene with a splash of color in the light after the rain. Our result captures the colorful and light atmosphere while retaining the structure of the house very well. In contrast, other methods, such as StyleFormer and AdaAttN, fail to reflect the colorful glowing effect.

**Quantitative Evaluation.** The Table 1 presents the DISTS values (Ding et al., 2020) of the output images compared to the target images after undergoing style transfer through various methods. The

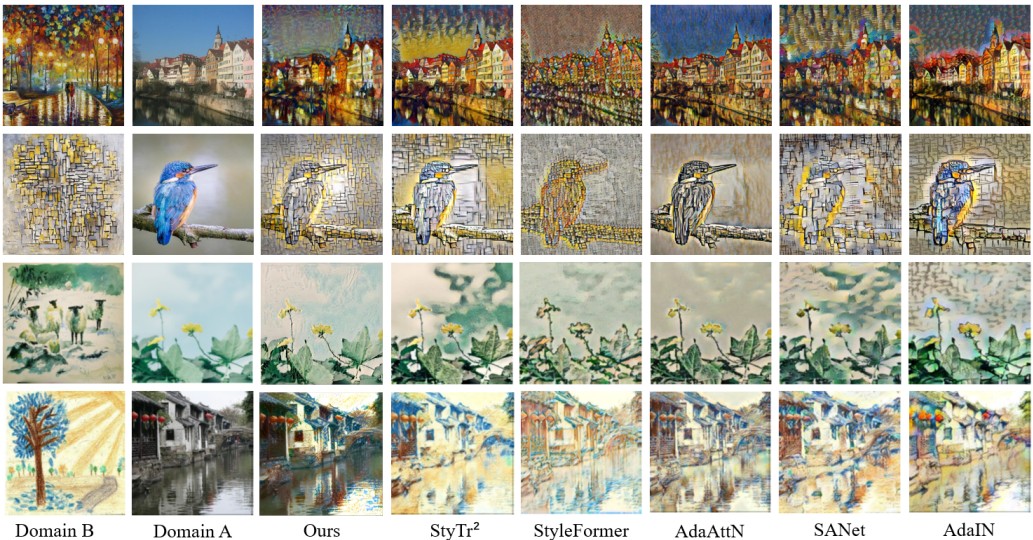

Figure 3: The comparison of our model against several famous I2I style rendering approaches.

Table 1: Quantitative comparison of content and style DISTS value with other approaches

|  | Ours | StyTr$^2$ | AdaAttN | AdaAttN | SANet | AdaIN |
|---|---|---|---|---|---|---|
| **sample_1** | | | | | | |
| $D_S$ | **0.2643**[1] | **0.2685**[2] | 0.3574 | 0.3244 | 0.2709 | 0.2871 |
| $D_C$ | **0.4251**[1] | 0.4384 | 0.4745 | 0.4423 | 0.4569 | **0.4308**[2] |
| **sum** | **0.6894**[1] | **0.7069**[2] | 0.8319 | 0.7667 | 0.7278 | 0.7179 |
| **sample_2** | | | | | | |
| $D_S$ | **0.2482**[1] | **0.2803**[2] | 0.3492 | 0.3832 | 0.2879 | 0.3327 |
| $D_C$ | 0.4522 | 0.4531 | 0.5285 | **0.3614**[1] | 0.4864 | **0.4442**[2] |
| **sum** | **0.7004**[1] | **0.7334**[2] | 0.8777 | 0.7446 | 0.7743 | 0.7769 |
| **sample_3** | | | | | | |
| $D_S$ | **0.3013**[2] | 0.3121 | 0.3160 | 0.3016 | **0.2938**[2] | 0.3394 |
| $D_C$ | **0.2780**[1] | 0.3311 | **0.3310**[2] | 0.3293 | 0.3510 | 0.3686 |
| **sum** | **0.5793**[1] | 0.6432 | 0.6470 | **0.6309**[2] | 0.6448 | 0.7080 |
| **sample_4** | | | | | | |
| $D_S$ | 0.4009 | **0.2969**[2] | 0.3222 | 0.3415 | **0.2857**[1] | 0.3389 |
| $D_C$ | **0.3902**[1] | 0.4940 | 0.5674 | **0.4672**[2] | 0.5214 | 0.4848 |
| **sum** | **0.7911**[2] | **0.7909**[1] | 0.8896 | 0.8087 | 0.8071 | 0.8237 |

table indicates the distance to both the content and style images from two distinct domains, represented by $D_S$ and $D_C$, respectively. The top-performing scores, as indicated by being ranked first or second, are highlighted in **bolded** and denoted with superscripts [1] and [2], respectively. The results presented in the Table 1 demonstrate that our proposed model consistently ranks first or second in terms of performance. Notably, the score for $S_C$ demonstrates that our model is particularly effective in preserving semantic content and consistently ranks first in this category. This suggests that our model is able to effectively stylize images while maintaining valid content features from the images in Domain A, resulting in outstanding semantic recognizability performance.

Furthermore, our model's unique conditioning mechanism allows for the degree of texture rendering to be progressively modified over semantic structure preservation, providing greater flexibility in terms of stylistic results. More details can be found in Appendix C and D. By adjusting the Control

Table 2: Quantitative comparison of user study

|  | Ours | StyTr$^2$ | AdaAttN | SANet | AdaIN |
|---|---|---|---|---|---|
| **General** | **4.16**[1] | **4.08**[2] | 3.69 | 3.03 | 3.13 |
| **Texture** | **4.01**[1] | **3.98**[2] | 3.16 | 3.56 | 3.28 |
| **Structure** | **4.34**[1] | 4.09 | **4.16**[2] | 3.03 | 3.41 |

Factor (CF) $\phi$, the trade-off between structure and texture can be freely modified according to the user's personal needs. The effect of $\phi$ on style transfer is further examined in Appendix B.

## 4.4 USER STUDY

To further evaluate the performance of our proposed method, we conducted a user study comparing it to several established baselines: StyTr2 (Deng et al., 2022), AdaAttN (Liu et al., 2021), SANet (Park & Lee, 2019), and (Huang & Belongie, 2017). We designed our questionnaire based on the work of (Li & Chen, 2009) and the specific options definition can be found in Appendix G. The questionnaire options included: A. General: the overall stylization of the image; B. Texture: the accuracy of the imitation of texture strokes; C. Structure: the preservation of the content structure after stylization. A total of 117 participants, including 56 males and 61 females with a diverse ethnic background, participated in the study. The participants' professional backgrounds included art workers, computer science researchers, and the general public. We used the images from section 4.3 for comparison and 20 images were randomly selected for each participant to rate. The minimum time for each marking was 20 seconds. The results were recorded in Table 2, with the top-performing scores highlighted in **bolded** and denoted with superscripts [1] and [2], respectively. The results in Table 2 clearly demonstrate that our method outperforms the other methods in terms of overall style rendering, imitation of stylized brushstroke textures, and preservation of semantic structure. In particular, our method excels in preserving semantic structures in the style rendering process.

**More Discussion.** Based on the experimental results, we can effectively control the degree of stylization in the output images using $\phi$ to meet the requirements of different scenarios. Moreover, in comparison to other baseline methods, our model exhibits outstanding performance in preserving the semantic structure of the content images while achieving style transfer, which is also validated in Table 1 and 2. In Appendix E, we demonstrated the rationality of the parameters and components in our algorithm via the **ablation study**.

## 5 CONCLUSION

In this work, we proposed a novel Style-Diffusion method for image-to-image (I2I) style transfer. Our proposed method provides a new approach to estimate the drift of the inverse process SDE using prior feature distributions extracted from two source domains. To the best of our knowledge, this is the first study to investigate how to use diffusion models to implement image style diffusion towards Schrödinger Bridge Problem, achieving impressive stylization while extraordinarily preserving the semantic structure of the source image. With Control Factor $\phi$, the degree of stylization in images can be adjusted according to different task requirements. Furthermore, we formulate the original muti-end diffusion problem under the perspective of a Schrödinger half bridge composite Problem, which not only reduces the computational cost of solving, but also provides a method for SGM to deal with the diffusion process of multiple complex distributions.

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

# A  THE SCHRÖDINGER BRIDGE PROBLEM

In this section, we will provide a detailed solution for the two problems proposed in Section 3.5 of the original article.

**The First Process.** In the $1^{st}$ process of our approach, we can rewrite it though the perspective of SBP:

**Problem 1.** *The $1^{st}$ process is to find a distribution from $\mathcal{D}(\alpha_0, \alpha_{N/2})$ that minimizes the KL-divergence: $\mathbb{Q}^{1*} := \arg\min\{D_{KL}(\mathbb{Q}^1\|\mathbb{W}^1) \mid \mathbb{Q}^1 \in \mathcal{D}(\alpha_0, \alpha_{N/2})\}$, where $\alpha_0 = p_d(X_A)$, $\alpha_{N/2} = p_\theta(v)$, $\mathbb{W}^1$ is a prior reference measure.*

*Proof.* Let $\mathbb{D}$ denote the set of all probability measures on $\Omega$ which are absolutely continuous with respect to stationary Winener measure. By Girsanov's theorem any $\Pi \in \mathbb{D}$ has a forward drift ($\mu(t)$), and a backward drift ($\lambda(t)$), the canonical process has Itô differential such that:

$$\text{forward: } dx(t) = \mu(t)dt + d\mathcal{W}^+(t) \tag{19}$$

$$\text{backward: } dx(t) = \lambda(t)dt + d\mathcal{W}^-(t) \tag{20}$$

where $\mathcal{W}^+(t), \mathcal{W}^-(t)$ are standard Wiener processes adapted to the forward and reverse time diffusion. By defining $b(t, x(t)) = \mu(t) - \nabla \ln \phi_t(x)$ (Pavon & Wakolbinger, 1991), where $\phi_t \cdot \hat{\phi}_t = q_t$ and $q_t$ represents the density of $x(t)$ that satisfies the Fokker-Planck (FPK) equation for the process of the form $dx(t) = b(t, x(t))dt + d\mathcal{W}(t)$, and referring to *e.g.* (Pavon & Wakolbinger, 1991) (Lemma 3.8) and (Léonard, 2014) (Theorem 2.4), the KLD between $\mathbb{Q}^1$ and $\mathbb{W}^1$ can be expressed as a decomposition:

$$D_{KL}[\mathbb{Q}^1\|\mathbb{W}^1] = \overbrace{D_{KL}[\mathbb{Q}_0^1\|\mathbb{W}_0^1]}^{constant} \\ + \mathbb{E}_{\mathbb{Q}^1}\left[\int_0^{\frac{N}{2}} \frac{1}{2}\|\mu(t) - b(t, x(t))\|^2 d(t)\right] \tag{21}$$

where $\mathbb{Q}_0^1$ and $\mathbb{W}_0^1$ denote the initial densities of $\mathbb{Q}^1$ and $\mathbb{W}^1$, and the first term is constant. By the Theorem 3.9 (Pavon & Wakolbinger, 1991), we can obtain the forward equivalent objective for SBP of the $1^s t$ process such that:

$$F(\mathbb{Q}^1) := \min_{\mathbb{Q}^1 \in \mathcal{D}(\alpha_0, \alpha_{N/2})} \mathbb{E}_{\mathbb{Q}^1}\left[\int_0^{\frac{N}{2}} \frac{1}{2}\|\mu(t) - b(t, x(t))\|^2 d(t)\right] \tag{22}$$

Using reverse diffusion, we can also obtain the backward equivalent objective for SBP of the $1^{st}$ process:

$$B(\mathbb{Q}^1) := \min_{\mathbb{Q}^1 \in \mathcal{D}(\alpha_0, \alpha_{N/2})} \mathbb{E}_{\mathbb{Q}^1}\left[\int_0^{\frac{N}{2}} \frac{1}{2}\|\lambda(t) - b_-(t, x(t))\|^2 d(t)\right] \tag{23}$$

Then, there holds:

$$D_{KL}[\mathbb{Q}^1\|\mathbb{W}^1] = D_{KL}[\mathbb{Q}_0^1\|\mathbb{W}_0^1] + F(\mathbb{Q}^1) \\ = D_{KL}[\mathbb{Q}_{N/2}^1\|\mathbb{W}_{N/2}^1] + B(\mathbb{Q}^1) \tag{24}$$

where $\mathbb{Q}_{N/2}^1$ and $\mathbb{W}_{N/2}^1$ denote the initial densities of $\mathbb{Q}^1$ and $\mathbb{W}^1$ at time index $t = \frac{N}{2}$.

**Half Bridge Problem.** To simplify the numerical solution of the iterative algorithms, we set $\alpha_0$ as initial value and force it into single-constraint problems, which transforms the original problem into a half bridge problem (Pavon et al., 2021). Then the forward and backward half bridge of the $1^{st}$ process is given by:

$$\text{forward: } \mathbb{Q}^{1*} = \inf_{\mathbb{Q}^1 \in \mathcal{D}(\alpha_0, \cdot)} D_{KL}(\mathbb{Q}^1\|\mathbb{W}^1)$$

$$\text{backward: } \mathbb{P}^{1*} = \inf_{\mathbb{P}^1 \in \mathcal{D}(\cdot, \alpha_{N/2})} D_{KL}(\mathbb{P}^1\|\mathbb{W}^1)$$

Using *e.g.* (Pavon et al., 2021) and (Vargas, 2021)(Theorem 9&10), the optimal solution of static forward bridge holds:

$$q^*(x, y) = p^{\mathbb{W}}(x, y)\frac{\alpha_0(x)}{p^{\mathbb{W}}(x)} \tag{25}$$

where $p^{\mathbb{W}}(x,y) = p_0^{\mathbb{W}}(x)p^{\mathbb{W}}(y|x)$ with marginal prior $p_0^{\mathbb{W}}(x)$, the joint distribution $q(x,y) \in \mathcal{D}(\alpha_0(x), \alpha_{N/2}(y))$, $\alpha_0(x) = \int q(x,y)dy$, $\alpha_{N/2}(y) = \int p(x,y)dx$. The optimal solution of static backward bridge holds:

$$p^*(x,y) = p^{\mathbb{W}}(x,y)\frac{\alpha_{N/2}(y)}{p^{\mathbb{W}}(y)} \tag{26}$$

Now, we have completed the description of the $1^{st}$ process from the perspective of SBP and provided the objectives. Half bridge's solutions can be considered "closed-form" to some extent, they can also be used to remove constraints by including them as an initial value problem, which provides simplification objectives for solving SBPs problems using iterative methods.

**The Second Process.** Similar to the $1^{st}$ process, the $2^{nd}$ process can also be represented as a description in terms of SBP.

**Problem 2.** *The $2^{nd}$ process is to find a distribution from $\mathcal{D}(\alpha_{N/2}, \alpha_N)$ that minimizes the KL-divergence: $\mathbb{Q}^{2*} := \arg\min\{D_{KL}(\mathbb{Q}^2\|\mathbb{W}^2) \mid \mathbb{Q}^2 \in \mathcal{D}(\alpha_{N/2}, \alpha_N)\}$, where $\alpha_{N/2} = p_\theta(v)$, $\alpha_N = p_d(X_B)$, $\mathbb{W}^2$ is a prior reference measure.*

Similar to the discussion of the $1^{st}$ process, the forward and backward half bridge of the $2^{nd}$ process is given by:

$$\textit{forward: } \mathbb{Q}^{2*} = \inf_{\mathbb{Q}^2 \in \mathcal{D}(\alpha_{N/2}, \cdot)} D_{KL}(\mathbb{Q}^2\|\mathbb{W}^2)$$

$$\textit{backward: } \mathbb{P}^{2*} = \inf_{\mathbb{P}^2 \in \mathcal{D}(\cdot, \alpha_N)} D_{KL}(\mathbb{P}^2\|\mathbb{W}^2)$$

So far, the formulation of the $2^{nd}$ process towards SBP and the objectives are given.

# B CONTROL FACTOR

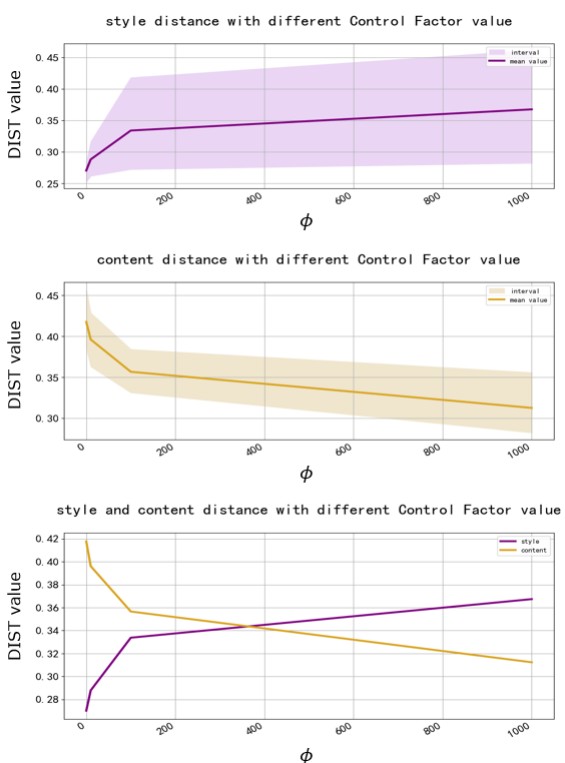

Figure 4: The DIST value with different control factor $\phi$.

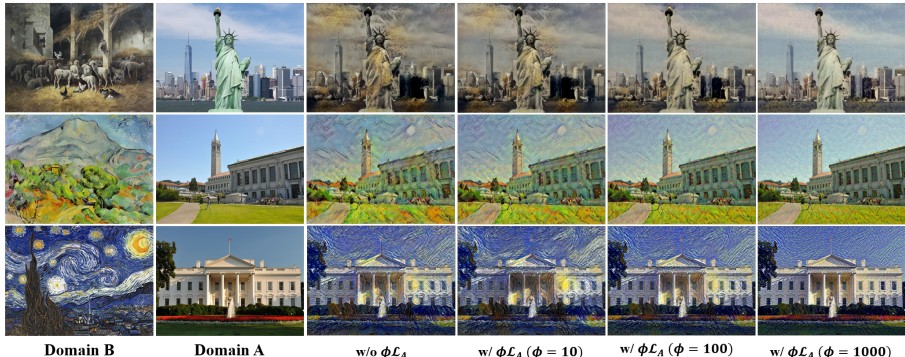

Figure 5: The outputs with different control factor $\phi$.

Table 3: DISTS value with different $\phi$.

|  | $\phi = 0$ | $\phi = 10$ | $\phi = 100$ | $\phi = 1000$ |
|---|---|---|---|---|
| sample_1 | | | | |
| $D_S$ | 0.2825 | 0.3161 | 0.4179 | 0.4613 |
| $D_C$ | 0.4574 | 0.4286 | 0.3544 | 0.2994 |
| sample_2 | | | | |
| $D_S$ | 0.2760 | 0.2859 | 0.3111 | 0.3424 |
| $D_C$ | 0.3859 | 0.3630 | 0.3312 | 0.2822 |
| sample_3 | | | | |
| $D_S$ | 0.2521 | 0.2617 | 0.2723 | 0.2986 |
| $D_C$ | 0.4098 | 0.3969 | 0.3841 | 0.3558 |

The distinct characteristic of our model in comparison to other methods is the conditioning mechanism, which allows for the flexibility to modify the degree of texture rendering while maintaining semantic structure preservation. Our model can produce a wide range of stylized results by adjusting the Control Factor (CF), $\phi$, to control the balance between structure and texture, as illustrated in Figure 5.

**Qualitative evaluation.** By altering the value of $\phi$, the level of stylization and the semantic structure in the generated image can be adjusted. As depicted in Figure 5, as the value of $\phi$ increases, the semantic structure of the image becomes more defined (e.g. windows and doors on buildings, outlines of statues, etc.), while at the same time the level of stylization decreases (e.g. brushstrokes and textures in the image, etc.). It is important to note that one cannot excessively reduce the value of $\phi$ in an effort to achieve a stronger stylistic transition, as this can result in certain areas of the image becoming overwhelmed (e.g. when $\phi = 1$ and 10 in sample 1 and 2). Similarly, the value of $\phi$ should not be increased excessively in an attempt to obtain a sharper semantic structure, as this can result in the generated image being insufficiently stylized (e.g. the stylized strokes and textures from Domain B are weak at $\phi = 1000$ in samples 1 and 3).

**Quantitative evaluation.** Table 3 records the DISTS values (Ding et al., 2020) of the generated images in Figure 5 in comparison to the target style image and the original content image. This data can then be used to generate the line graph depicted in Figure 4. Through quantitative analysis, it is evident that there is a clear trade-off between style DIST and content DIST, indicating that an enhancement in stylization is accompanied by a loss of semantic structural information. Furthermore, it can be observed that as the semantic structure becomes sharper, the stylization is weakened.

## C  PROGRESSIVE RENDERING IMPLEMENTATION

As described in the original text, we employed VGG as an encoder to extract corresponding feature from the content and style images as priors. The content feature target vector is extracted from

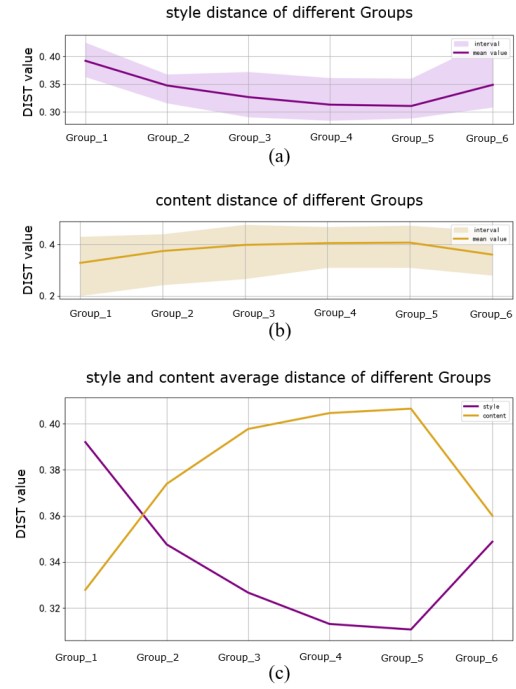

Figure 6: The DISTS value varies over different Groups.

layer 'conv4_2', and the style feature target vectors are extracted from layers 'conv1_1', 'conv2_1', 'conv3_1', 'conv4_1' and 'conv5_1' (Gatys et al., 2016). In this section, we will discuss the impact of using varying numbers of features on the final output results.

**Group Design.** In Group_1, we only utilized the style feature ('conv1_1') from the style image. In Group_2, we employed style features ('conv1_1' and 'conv2_1') from the style image. In Group_3, we utilized style features ('conv1_1', 'conv2_1' and 'conv3_1') from the style image. In Group_4, we employed style features ('conv1_1', 'conv2_1', 'conv3_1' and 'conv4_1') from the style image. In Group_5, we utilized style features ('conv1_1', 'conv2_1', 'conv3_1', 'conv4_1' and 'conv5_1') from the style image. In Group_6, we employed style features ('conv1_1', 'conv2_1', 'conv3_1', 'conv4_1' and 'conv5_1') from the style image and content feature ('conv4_2') from the content image. Furthermore, we only impose the CF term in group 6 among all the groups.

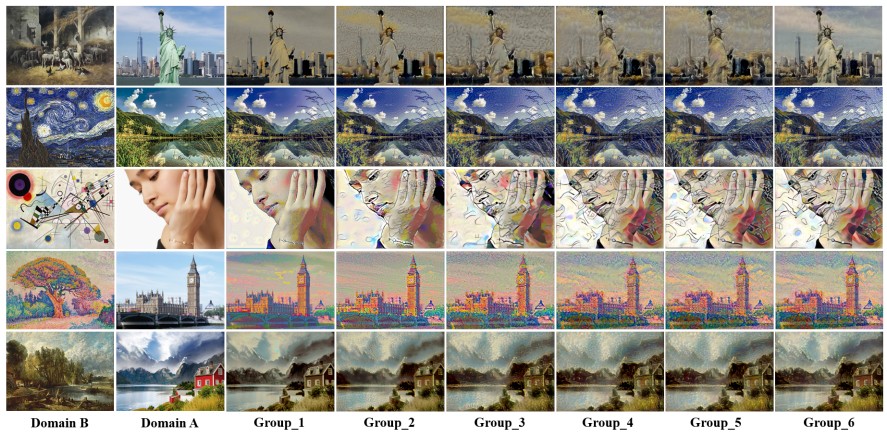

Figure 7: The progressive changes of each Group.

Table 4: Quantitative DISTS value with different Groups.

|  | Group_1 | Group_2 | Group_3 | Group_4 | Group_5 | Group_6 |
|---|---|---|---|---|---|---|
| sample_1 | | | | | | |
| $D_S$ | 0.3892 | 0.3671 | 0.3438 | 0.3444 | 0.3323 | 0.4309 |
| $D_C$ | 0.3673 | 0.3968 | 0.4043 | 0.3928 | 0.3939 | 0.3325 |
| sample_2 | | | | | | |
| $D_S$ | 0.3788 | 0.3312 | 0.3059 | 0.2654 | 0.2637 | 0.2817 |
| $D_C$ | 0.2016 | 0.2437 | 0.2674 | 0.3108 | 0.3109 | 0.2810 |
| sample_3 | | | | | | |
| $D_S$ | 0.4041 | 0.3571 | 0.3206 | 0.3101 | 0.3086 | 0.3089 |
| $D_C$ | 0.2989 | 0.3573 | 0.3952 | 0.4123 | 0.4039 | 0.3889 |
| sample_4 | | | | | | |
| $D_S$ | 0.4242 | 0.3661 | 0.3716 | 0.3607 | 0.3597 | 0.3663 |
| $D_C$ | 0.3433 | 0.4333 | 0.4748 | 0.4662 | 0.4716 | 0.4492 |
| sample_5 | | | | | | |
| $D_S$ | 0.3637 | 0.3165 | 0.2914 | 0.2851 | 0.2892 | 0.3561 |
| $D_C$ | 0.3673 | 0.3968 | 0.4043 | 0.3928 | 0.3939 | 0.3325 |
| Average | | | | | | |
| $D_S$ | 0.3920 | 0.3476 | 0.3267 | 0.3131 | 0.3107 | 0.3488 |
| $D_C$ | 0.3279 | 0.3739 | 0.3977 | 0.4046 | 0.4065 | 0.3601 |

**Qualitative evaluation.** As shown in Figure 7, as the number of style feature targets imposed on the images increases from Group_1 to Group_5, the stylization in the images becomes more prominent (e.g. brush strokes and textures). However, we can observe that simply increasing the style features could result in certain critical semantic structures in the image becoming increasingly blurred. With the reference of $\phi$ and the content feature target, the images in Group_6 not only successfully achieve stylization but also make the semantic structures clearer than the previous outputs.

**Quantitative evaluation.** The Table 4 presents the DISTS values (Ding et al., 2020) of the output images compared to the target images, and it can be observed that as more style feature targets are progressively imposed on each Group, the style DISTS value is getting smaller, indicating that the generated images increasingly closely resemble the target style images in terms of texture and style. In contrast, the content DISTS value is increasing, which indicates that the generated images lose more and more information about their semantic structure as they are stylized, resulting in a gradual blurring of contours from the content image. However, with the guidance of the content feature target and the CF $\phi$ in Group_6, the content DISTS value plummets, meaning that the image becomes more similar to the content image from Domain A in terms of semantic structure. At the same time, the style DISTS value becomes larger, indicating a drop in performance at the stylized level compared to the previous Group.

This analysis and Figure 6 reveal that in the process of style transfer, there is a trade-off between preserving semantic structure and stylization, and it is difficult to preserve both at the same time. However, our model's unique CF mechanism allows for greater flexibility in terms of controlling the extent to which semantic structure is preserved, enabling the user to select results from Group_6 when a clearer content image is desired, or results from Group_5 when stronger stylization is needed.

## D  SEMANTIC STRUCTURE PRESERVATION

When performing style transfer from a content image to an abstract painting, it particularly tests the model's ability to preserve semantic structures. As shown in Figure 8, our model exhibits strong performance in preserving image semantic structures. In the original image, the woman is wearing a bracelet on her wrist, which is not present in the rendered result in Group_3 due to the absence of the content target or the $\phi$ that adjusts the sharpness of the original semantic structure in the output image. Furthermore, because Group_3 only imposes lower-level vectors ('conv1_1', 'conv2_1' and

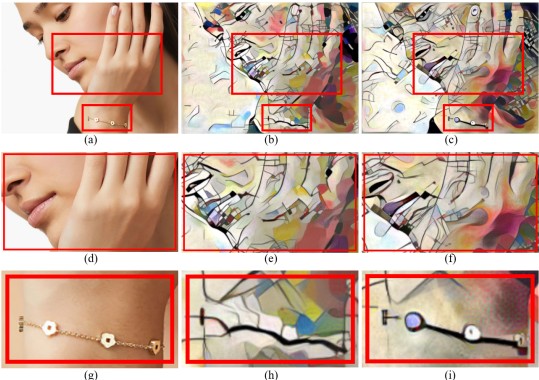

Figure 8: Evaluation of semantic structure preservation. The original image is denoted by (a), the image from Group_3 is denoted by (b), and the image from Group_6 is denoted by (c). Detailed display images corresponding to (d) through (i) are also provided.

'conv3_1') of CNN, it is more biased towards simulating local small structure textures from Domain B, resulting in a cluster of small structures near the nose and mouth. In contrast, Group_6 imposes more high-level features ('conv4_1' and 'conv5_1'), the content target and $\phi$, which controls the extent of semantic structure preservation. As a result, the bracelet and the hand curve, two semantic structures in the source image, are nicely preserved in the final rendering results in Group_6, further demonstrating the superior performance of our method.

## E   ABLATION STUDY

In this section, we conduct several ablation studies on the number of reference feature vectors $\{\sum_{i=1}^{5}(s_i, c_1)\}$, the distance of Metric Space $L_D$, loss term and the control factor (CF) $\phi$.

**Reference Feature Vectors.** In the experiment, by utilizing Equation (14) (Section 3.3) as our loss function, setting $\phi$ to 358, and using the Euclidean distance $L_D$, we impose a varying number of $\{\sum_{i=1}^{5}(s_i, c_1)\}$ as prior references and obtain different output images, as shown in Figure 9. Based on the outputs' performance, with a small number of style features as references, the style

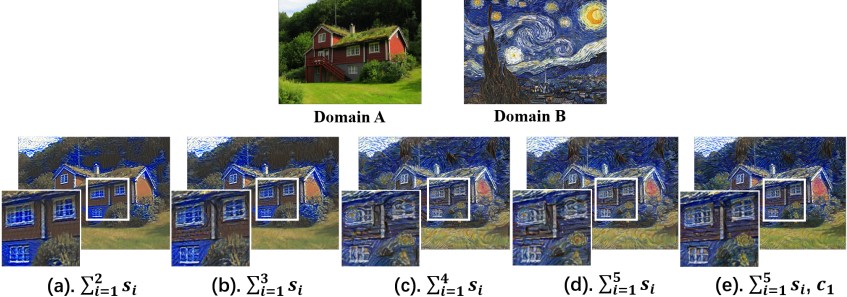

Figure 9: Ablation study of different number of target feature vectors. More detailed results and evolution are illustrated in supplementary materials.

transfer results are not satisfactory. With more style features without content feature, the stylization is improved, but the semantic structure is not well preserved. With all content and style features, the output image can preserve clear semantic content structure and achieve extraordinary style transfer.

**Metric Space Distance.** In order to verify the effectiveness of distances in different metric spaces on model performance, we compared the images generated using the three distances defined in Section 3.3, as shown in Figure 10. The results demonstrate that using the $\mathbb{C}$ *space* cannot complete the style transfer task smoothly; using the $l^p$ *space* results in poorer style transfer effect, and some semantic

content is not achieved in style transfer (as shown in the red box in the figure); using the *Euclidean space* $\mathbb{R}^n$ can achieve satisfactory performance.

**Loss Term and the Control Factor.** To validate the necessity and rationality of the control factor term $\phi\mathcal{L}_A$ in Equation (14) (Section 3.3), we conducted comparative experiments as shown in Figure 11. In the experiments, we observed the influence of the $\phi\mathcal{L}_A$ on the experimental results, and obtained corresponding output images by modifying the hyperparameter $\phi$. The results show that when the $\phi\mathcal{L}_A$ is not introduced, or the $\phi$ value is small, the style transfer will cause the semantic structure of the content image to be blurred and lose some semantic content. Moreover, serious overflow will occur in the originally clean background. Increasing $\phi$ will alleviate the above problems, but if $\phi$ is too large, the degree of stylization will be low, affecting the effect of style transfer.

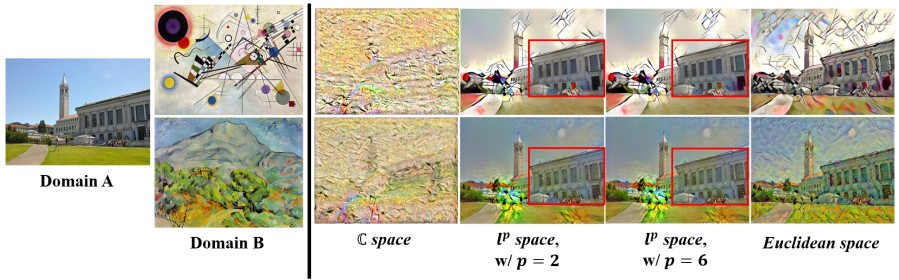

Figure 10: Ablation study of the Metric space distance.

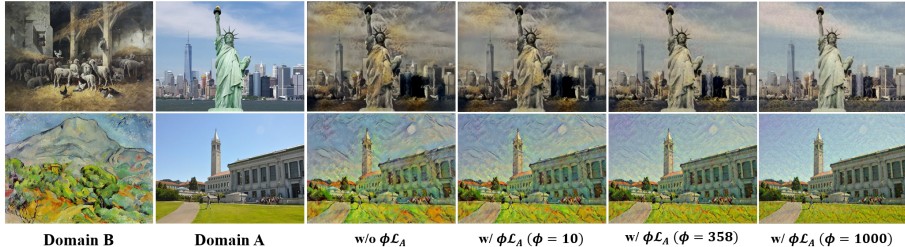

Figure 11: Ablation study of the Loss term and the control factor. We compared the results generated by using or not using the CF term in loss function, and observed the impact of adjusting the $\phi$ value on the images. More quantitative analysis and further discussions on this are presented in the supplementary material.

## F    EXTRA TEST SAMPLES

Due to the page limit, we only showcased four samples in Figure 3 in the paper. Figure 12 shows the remaining samples used in our tests.

## G    USER STUDY

User study has been discussed in Section 4.4. In this section, we provide an example illustrated in Figure 13 to demonstrate the formatting of the questions and the description of the options in our questionnaire. The participants in this user study have been informed of the content and objectives of our research.

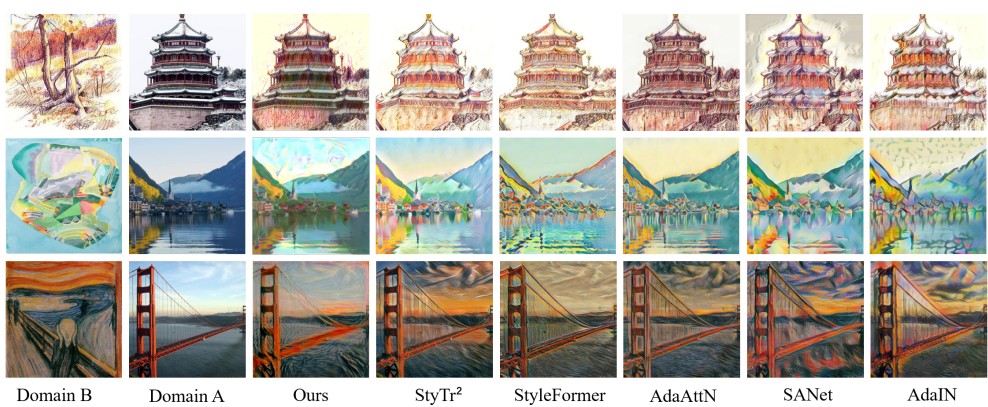

| Domain B | Domain A | Ours | StyTr² | StyleFormer | AdaAttN | SANet | AdaIN |

Figure 12: This figure presents the extra samples.

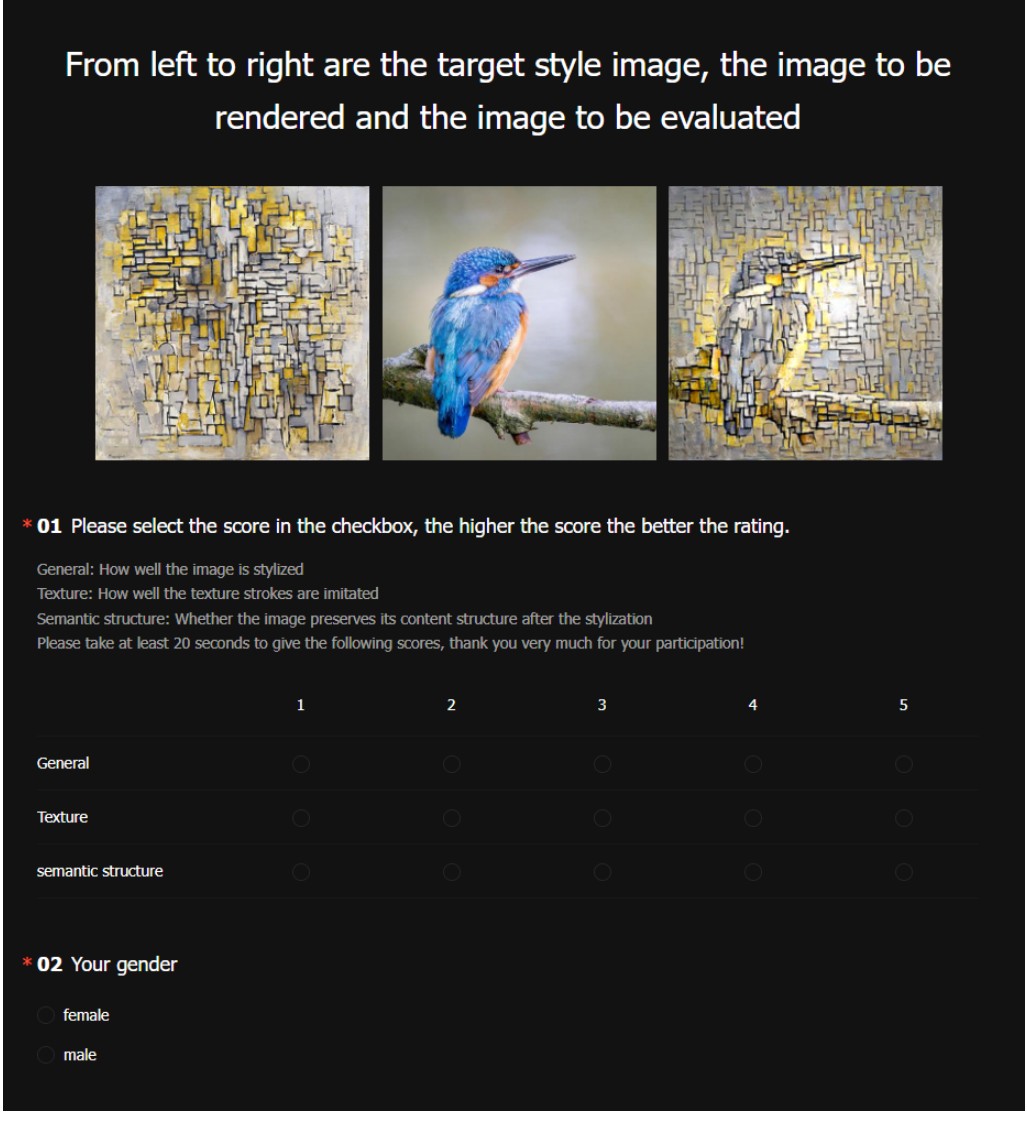

Figure 13: Example of questionnaire

