# OpenReview forum: "Delve into Image Style Diffusion Towards Schrödinger Bridge Problem"
_ICLR.cc/2024/Conference — ICLR 2024 Conference Withdrawn Submission_

### Official Review · Reviewer_p74s · 2023-10-29

**Soundness:** 2 fair
**Presentation:** 3 good
**Contribution:** 2 fair
**Rating:** 5
**Confidence:** 4

**Summary:**

This paper proposes to employ approximate score-matching based on style and content reference image during the reverse time for image style transfer. Besides, such a proposal provides a control factor to control the degree of stylization. The authors claimed this as the first work of using the diffusion model for style transfer. They show both quantitative and qualitative evaluations of MS-COCO and WikiArt.

**Strengths:**

1. The authors provide comprehensive math explanations and theorems in the paper for illustration.

2. This paper is well-written and it is easy to follow.

**Weaknesses:**

1. The authors claim this submission is the first work using the diffusion model for style transfer. However, I think there have been some works on this point. The authors should discuss and compare with them.


[1] Zhang Y, Huang N, Tang F, Huang H, Ma C, Dong W, Xu C. Inversion-based style transfer with diffusion models. In Proceedings of the IEEE/CVF Conference on Computer Vision and Pattern Recognition 2023 (pp. 10146-10156).

[2] Wang Z, Zhao L, Xing W. StyleDiffusion: Controllable Disentangled Style Transfer via Diffusion Models. In Proceedings of the IEEE/CVF International Conference on Computer Vision 2023 (pp. 7677-7689).

2\. The proposed method is not sound to me. Specifically, I have read sections 3.2 - 3.4 but still not clear which distance metric is used to calculate the loss function of content and style. Besides, during reverse, images and predicted images are typically added noises while the pre-trained CNN is trained with clean images, how do the authors overcome such limitations? What’s the relationship between the proposal and classifier guidance [1] or self-guidance [2]? The authors should discuss and compare with them.

[1] Dhariwal P, Nichol A. Diffusion models beat gans on image synthesis. Advances in neural information processing systems. 2021 Dec 6;34:8780-94.

[2] Epstein D, Jabri A, Poole B, Efros AA, Holynski A. Diffusion self-guidance for controllable image generation. arXiv preprint arXiv:2306.00986. 2023 Jun 1.

3\. The experiment results are not strong enough to verify the effectiveness of the proposal. Specifically, most results of StyTr^2 show better results in Figure 3 and Table 1.  For example, in the second sample in Figure 3, StyTr^2 shows a much clearer outline of the bird compared with the proposal. In the third sample in Figure 3, StyTr^2 renders a better cloud than the proposal. Besides, the authors should compare with the works mentioned in the first point.

**Questions:**

My biggest concern is about the comparison with related works. Please see more details in the Weaknesses.

---

### Official Review · Reviewer_zeKW · 2023-10-31

**Soundness:** 2 fair
**Presentation:** 1 poor
**Contribution:** 2 fair
**Rating:** 3
**Confidence:** 3

**Summary:**

This paper proposes Style-Diffusion which is a diffusion model that conducts image-to-image style transfer. Style-Diffusion is built upon Schrodinger Bridge problem, where they aim to conduct neural style transfer via sampling from the midst point of two different domains. To this end, in addition to the score matching objective, the author proposes to use the VGG features to condition the diffusion model, which helps preserving the semantic structure from the content image. The experiments were conducted to show its effectiveness.

**Strengths:**

Style-Diffusion incorporates VGG feature networks in image-to-image diffusion model to enhance style transfer without deforming semantic structure.

**Weaknesses:**

- Some experimental details are missing, for example, which diffusion models are used in an experiment? How the noise variances or timesteps are chosen? How does the sampling be done at the midst?
- The writing could be improved. For example, why do we need Figure 2 in this paper? It seems like Figure 2 simply depicts the conventional diffusion process, but does not provide any information on the proposed method. Also, in the related works section, the authors defined a score-based model, while in the methods section, the model suddenly changed into a noise prediction model.
- There are some missing references that considers image-to-image translation using diffusion models (or SGMs) such as DDIB [https://arxiv.org/abs/2203.08382] or I^2SB [https://arxiv.org/abs/2302.05872]. I believe the concurrent works on diffusion models with Schrodinger Bridges for image-to-image translation should be discussed even though they do not consider style transfer tasks.

**Questions:**

- See Weakness part. In general, my major concern is the lack of implementation details in training / sampling of the diffusion model. I think the idea of extracting features from pretrained vision encoders were widely used tactics in neural style transfer, thus the main contribution of this work lies in adapting to the diffusion models. However, many details are missing in terms of reproducibility.
- What is D3PSR in sentence below equation 15?

---

### Official Review · Reviewer_qGym · 2023-11-01

**Soundness:** 2 fair
**Presentation:** 1 poor
**Contribution:** 2 fair
**Rating:** 5
**Confidence:** 3

**Summary:**

This work introduces a Style-Diffusion method for stylization transfer in image generation tasks. The proposed method utilizes Score-Based Generative Modeling (SGM) and approximate score-matching to estimate the drift of the reverse-time Stochastic Differential Equation (SDE). By introducing the Control Factor, controllable stylization in the output images is achieved. The original diffusion problem is reformulated as a composite Schrödinger half bridge Problem to improve computation speed and enable diffusion evolution between more complex multiple distributions.

**Strengths:**

- It is interesting that the proposed method gradually evolves content and style image distributions (by utilizing the middle output), and introduces Control Factor (CF) to enable control over the degree of stylization in the output images.

- By adjusting the Control Factor (CF), the balance between structure and texture can be controlled.

- The original multi-end diffusion problem is re-formulated as a composite Schrödinger half bridge Problem to improve computation speed.

**Weaknesses:**

- It is not ture that this is the first work on image style transfer using diffusion models, please check the following paper:
StyleDiffusion: Controllable Disentangled Style Transfer via Diffusion Models, ICCV 2023
- The experiments have limitations as they lack comparison to the latest works on style transfer and the compared methods are relatively old. Additionally, the comparison to StyleDiffusion, which was mentioned above, is missing.
- The writing could be further improved, I found the paper somewhat difficult to follow.

**Questions:**

- The interpretation (page 6, the first three lines) of eq.13 is confusing to me. Shouldn't the first term represent the loss related to content, while the second term represents the loss related to style?

- There is a text description "The second and third terms denote ..." below eq. 14. However, it is confusing that there is no such term in the equation. Did I miss something obvious?

- What is the architecture used to implement the CNN for feature extraction? And what "conv1_1", "conv2_1" and so on stands for? And it is unclear how this CNN could be used as the noise predictor.

---

### Official Review · Reviewer_7QnQ · 2023-11-02

**Soundness:** 1 poor
**Presentation:** 1 poor
**Contribution:** 1 poor
**Rating:** 1
**Confidence:** 5

**Summary:**

This paper uses formulates style transfer as Schrodinger bridge problem.
- The score matching losses are defined in the feature space of an encoder by feeding x_t: the x_t should have the features of the content image and the style image.

The authors argue that the proposed method is flexible and efficient while preserving the semantic structure.

The authors argue that the proposed method is the first style transfer with a diffusion model.

The experiments are conducted with a few artistic style transfer examples.

**Strengths:**

Score matching in the VGG feature space is interesting.

**Weaknesses:**

1. Statements are unclear or not self-contained. Thorough revision with an expert would make the paper readable. The details are deferred to the end because there are too many.

2. Connection between Schrödinger bridge and style transfer is missing.
* What are the advantages of using Schrödinger formulation on style transfer? The authors mention reducing time cost and it is a general advantage in diffusion models without connection to style transfer.

3. Important recent papers with diffusion-based style transfer are not covered nor compared. Here are a few.
* Ruiz et al., Dreambooth, CVPR 2023
* Gal et al., An Image is Worth One Word, ICLR 2023

4. Qualitative evaluation is hardly agreeable.
* I would say the proposed method does not reflect the style in Sample 3.
* All methods produce colorful results without glowing effect.

5. Quantitative results are measured with too few (=4) images. Previous papers use hundreds, e.g., in StyTr2 and StyleFormer.

6. Quantitative results are measured with a non-standard metric.

-------------------
Here are details of the unclear and non self-contained statements.

* In what aspect the proposed method is flexible and efficient? Flexibility and efficiency are mentioned in the abstract and introduction sections but they should be more rigorously / specifically defined.

* What are prior distributions and domains? Figure 1 has boxes with domains and images but it is vague. Please add rigorous definition of the domains.

* Why is it important to estimate the *time inhomogeneous* drift?

* Definition of $X$ and $\mathcal{X}$ are missing.

* Hyperparameter $\phi$ is introduced in Section 3.2 but used in 3.3. Please move it close to the usage.

* Eq.11 should have $x_B$ instead of $x_A$.

* Section 3.2 and 3.3 are poorly structured.

> Existing diffusion-based methods leverage the U-Net architecture to obtain a noise predictor, $\epsilon_t^\theta$.
* U-Net is the noise predictor itself, not a separate component to obtain a noise predictor.

> Existing diffusion-based methods leverage the U-Net architecture to obtain a noise predictor, $\epsilon_t^\theta$.
> Different from the noise adding way of traditional diffusion model, which utilizes the Gaussian noise during the forward process. We impose the target vectors to guide the noise in each unique time step.
> ... we use a pre-trained CNN to extract 5 style feature target vectors (s1, s2, s3, s4, s5) and 1 content feature target vector (c1) from two domains, and then impose these feature targets to guide the noise, $\epsilon_t^\theta$.
* The first two sentences imply that the proposed method does not use U-Net but the last sentence explains that feature vectors from the CNN-based encoders guide the noise $\epsilon$ where $\epsilon$ is U-Net. Please resolve this contradiction. If I have mistaken it, please provide clear explanation.

> there is currently no work that utilizes a diffusion-based model for style transfer,
* Ruiz et al., Dreambooth, CVPR 2023
* Gal et al., An Image is Worth One Word, ICLR 2023

I am stopping here although there are far more errors.

(minor)

typos

> In contrast to the traditional SGM method, our model obtains the desired output at the midst layer.

-> midst timestep?

> Different from ... during the forward process. We impose the target vectors to guide ....

-> different from ... process, we impose ....

Sample IDs are missing in Figure 3

**Questions:**

I think this paper needs a thorough upgrade to be submitted anywhere else.